# Effects of anesthetic management on persistent pain after breast cancer surgery

Yu Sakai[1]*, Ayako Kobayashi[1], Mariko Akata[1], Kentaro Tojo[1], Takahiro Mihara[1], Akimitsu Yamada[2], Kazutaka Narui[3], Sadatoshi Sugae[4], Nobuyasu Suganuma[5], Takahisa Goto[1], Tomoyuki Miyazaki[1,6]*

1 Department of Anesthesiology, Yokohama City University School of Medicine, Yokohama, Japan, 2 Department of Gastroenterological Surgery, Yokohama City University School of Medicine, Yokohama, Japan, 3 Department of Breast and Thyroid Surgery, Yokohama City University Medical Center, Yokohama, Japan, 4 Department of Breast Surgery, Fujisawa City Hospital, Fujisawa, Japan, 5 Department of Surgery, Yokohama City University School of Medicine, Yokohama, Japan, 6 Center for Promotion of Research and Industry-Academic Collaboration, Yokohama City University, Yokohama, Japan

* ysakai61@yokohama-cu.ac.jp (YS); johney@yokohama-cu.ac.jp (TM)

## Abstract

### Background

Persistent pain after breast cancer surgery (PPBCS) affects 20–35% of patients, significantly impacting their quality of life. Although prevention through perioperative intervention is crucial, effective strategies to prevent PPBCS have not been established. In particular, the role of anesthetic management in preventing PPBCS remains controversial.

### Methods

This multicenter, retrospective, observational study included 183 women aged 20–70 years who underwent unilateral breast cancer surgery under general anesthesia between April 2012 and March 2014. Pain was assessed using a numerical rating scale (NRS) during follow-up visits. PPBCS was categorized as 'no' (NRS = 0), 'mild' (NRS = 1–2), and 'moderate-to-severe' (NRS ≥ 3) pain. Univariate and multivariate analyses evaluated associations between perioperative factors and PPBCS.

### Results

Of 183 participants, 127 (69.4%) reported PPBCS: 59 (32.2%) mild and 68 (37.2%) moderate-to-severe. No significant associations were found between anesthetic management factors (including total intravenous anesthesia vs. volatile anesthesia, intraoperative opioid doses, and use of adjuvant analgesics) and PPBCS incidence or intensity. Axillary lymph node dissection was significantly associated with moderate-to-severe PPBCS (odds ratio: 2.04; 95% confidence interval: 1.04–4.00).

**Data availability statement:** All relevant data are within the paper and its Supporting Information files.

**Funding:** This research was supported by JSPS KAKENHI Grant-in aid for Scientific Research (B) [No. 2323K27694] (TM), and partly supported by the grant for 2024-2026 Strategic Research Promotion (No. SK202404) of Yokohama City University (TM). The funders had no role in study design, data collection and analysis, decision to publish, or preparation of the manuscript.

**Competing interests:** The authors have declared that no competing interests exist.

## Conclusion

No significant associations were found between anesthetic management and PPBCS. Further research is needed to identify anesthetic factors that may prevent PPBCS.

---

## Introduction

Breast cancer is recognized as the most common type of cancer among women worldwide. More than 2.2 million new cases were diagnosed in 2020, accounting for 11.7% of all cancer diagnoses [1]. The primary treatment strategy for breast cancer is surgical intervention followed by chemotherapy and/or radiation therapy. Advances in early detection and therapeutic enhancements have markedly improved the survival rates of patients with breast cancer [2]. However, a significant proportion of patients experience persistent pain after breast cancer surgery (PPBCS). The incidence of PPBCS it reported to range between 20% and 35% [3–5], and its intensity is considerable, with a pooled pain score of 3.9 cm on a 10-cm visual analogue scale [5]. Notably, PPBCS not only causes pain at the surgical site but can also prolong physical and psychosocial dysfunction, markedly diminishing quality of life [6]. These findings highlight the need for targeted interventions to prevent and/or manage PPBCS.

The prevention of PPBCS through perioperative intervention is important because complete relief from PPBCS may be difficult to achieve once it occurs, due to a lack of clinical evidence to guide treatment decisions [7]. It is possible that specific anesthetic management may contribute to the prevention of PPBCS; however, there are several controversies surrounding this theory. Local anesthetic infiltration and paravertebral block reportedly decrease the incidence of PPBCS [8], while the efficacy of pectoralis nerve block in preventing PPBCS has not been determined [9,10]. Additionally, there are several conflicting reports on the effects of maintaining general anesthesia with inhalational agents or propofol on the prevention of PPBCS [11,12]. Furthermore, the effects of high-dose opioids administered during anesthesia, especially remifentanil, which can result in opioid-induced hyperalgesia (OIH) [13,14], on PPBCS remain unclear [12,15].

This study aimed to clarify the association of perioperative factors, including anesthetic management, with the subsequent incidence and intensity of PPBCS.

## Materials and methods

### Ethics statement

This study was conducted in accordance with the principles of the Declaration of Helsinki. The protocol for this study was reviewed and approved by the institutional review boards of the two participating hospitals. Written informed consent was obtained from all patients prior to the survey. This study was registered with the University Hospital Medical Information Network on the 8th of September 2014 (registration no. UMIN000015069).

## Study design and participants

This was a multicenter, retrospective, observational study. Patients who attended outpatient visits for postoperative follow-up at two hospitals were recruited based on the following inclusion criteria: women were ≥ 20 and < 70 years on the day of surgery, American Society of Anesthesiologists Physical Status 1–2, single unilateral breast cancer surgery (without subsequent reconstructive surgery) under general anesthesia at either of these two institutions between April 2012 and March 2014, and the outpatient visit was > 6 months after surgery. The exclusion criteria were as follows: body mass index (BMI) > 35, history of psychiatric disorders, preoperative opioid use, pre- and/or postoperative steroid use, other surgeries prior to participation in this study after breast cancer surgery, cancer recurrence or distant metastasis, other cancers, and death. The questionnaire-based survey was administered during outpatient follow-up visits at the hospitals.

## Data collection

A numerical rating scale (NRS) was used to determine the prevalence of PPBCS and the severity of patients' pain, with 0 indicating no pain and 10 indicating the worst pain imaginable [16,17]. After obtaining informed consent, participants were provided with the NRS questionnaire and responded independently regarding pain at their surgical site during the week prior to the survey by circling the number 1–10 they associated their pain with. The completed questionnaires were then returned to researchers during the outpatient visit. The participants' background information and physiological data were obtained from their electronic medical records. Medical records from April 2012 to March 2014 were accessed between September 2014 and March 2015 for research purposes. The authors had access to identifying information during data collection, after which all data were anonymized for analysis.

## Outcomes

In this study, a PPBCS of 0 on the NRS was defined as 'no,' 1–2 as 'mild,' and 3 or greater as 'moderate-to-severe.' This classification was based on the median NRS value of 2 in our study population and is consistent with previous literature suggesting that NRS scores of ≥3 often represent clinically significant pain requiring intervention [18]. The primary outcome was the association between the incidence of any intensity of PPBCS (NRS ≥ 1, 'mild' and 'moderate-to-severe') and perioperative factors. The secondary outcomes were as follows: association between the incidence of moderate-to-severe PPBCS and perioperative factors, and association between the intensity of PPBCS and perioperative factors.

## Statistical analyses

Normality of continuous variables was assessed by histogram analysis, and all continuous variables demonstrated non-normal distribution. All continuous variables are expressed as median and interquartile range (IQR), due to the lack of normality in distribution. Categorical variables are expressed as the number of cases and percentage of the total patients. Variables assumed to be related to PPBCS in previous studies were extracted [11,19], and the following 15 candidate variables were evaluated by univariate analysis: age (10-year increments), duration of postoperative course (months), BMI, axillary lymph node dissection (ALND), operative time, total intravenous anesthesia (TIVA) compared to volatile anesthesia, intraoperative fentanyl dosage (standardized by body weight), intraoperative remifentanil administration rate, surgical procedure, pectoral nerve (PECS) block, preoperative analgesic use (non-steroidal anti-inflammatory drugs [NSAIDs], acetaminophen, or gabapentinoids), pre- and/or postoperative chemotherapy, postoperative radiation therapy, and endocrine therapy. The variables to be analyzed in the multivariate analysis were discussed among a group of experienced anesthesiologists, and the following factors in the analysis were selected, taking into account risk factors described in previous papers and their clinical importance, were selected: TIVA compared to volatile anesthesia, intraoperative fentanyl dosage (standardized by body weight), intraoperative remifentanil administration rate, PECS block, ALND, age (10-year increments), and duration of postoperative course (months).

A logistic regression model was used to identify anesthetic factors associated with the incidence of PPBCS for both 'any pain' and 'moderate-to-severe pain'. Furthermore, to identify the association between anesthetic factors and the severity of PPBCS, multiple linear regression analysis was performed, with the NRS treated as a continuous variable. Multicollinearity was evaluated by calculating variance inflation factors. The data were analyzed using the Stata/IC 16 (Stata Corporation, College Station, TX, USA), with statistical significance set at $P < 0.05$.

## Results

A total of 302 patients attended postoperative outpatient follow-up visits at two institutions between September 2014 and March 2015. Among them, 252 met the inclusion criteria. Subsequently, 62 were excluded primarily due to cancer recurrence, metastasis, or psychiatric disorders (including the use of antidepressants or anxiolytic agents). Additionally, one patient declined to respond and six did not complete the questionnaire. Finally, 183 participants (96.3% of those who received the questionnaire) completed it and were included in our analyses (Fig 1). The participants' characteristics are listed in Table 1. Among the 183 participants, 56 (30.6%) reported 'no' pain (0 on the NRS) and 127 (69.4%) reported pain of 1 or more on the NRS, as follows: 59 (32.2%) reported 'mild' PPBCS (NRS 1–2) and 68 (37.2%) reported 'moderate-to-severe' (NRS ≥ 3). Patients underwent volatile anesthesia in 59.0% (sevoflurane [47.7%] and desflurane [11.4%]) and TIVA in 41.0% of the cases. Fentanyl and remifentanil were used in combination in most patients, and approximately half of the patients received remifentanil at an infusion rate of < 0.2 mcg/kg/min. The proportion of ALND was significantly higher in the 'moderate-to-severe' pain group than that in the 'no or mild' pain group. There were no other significant differences in characteristics between the two groups.

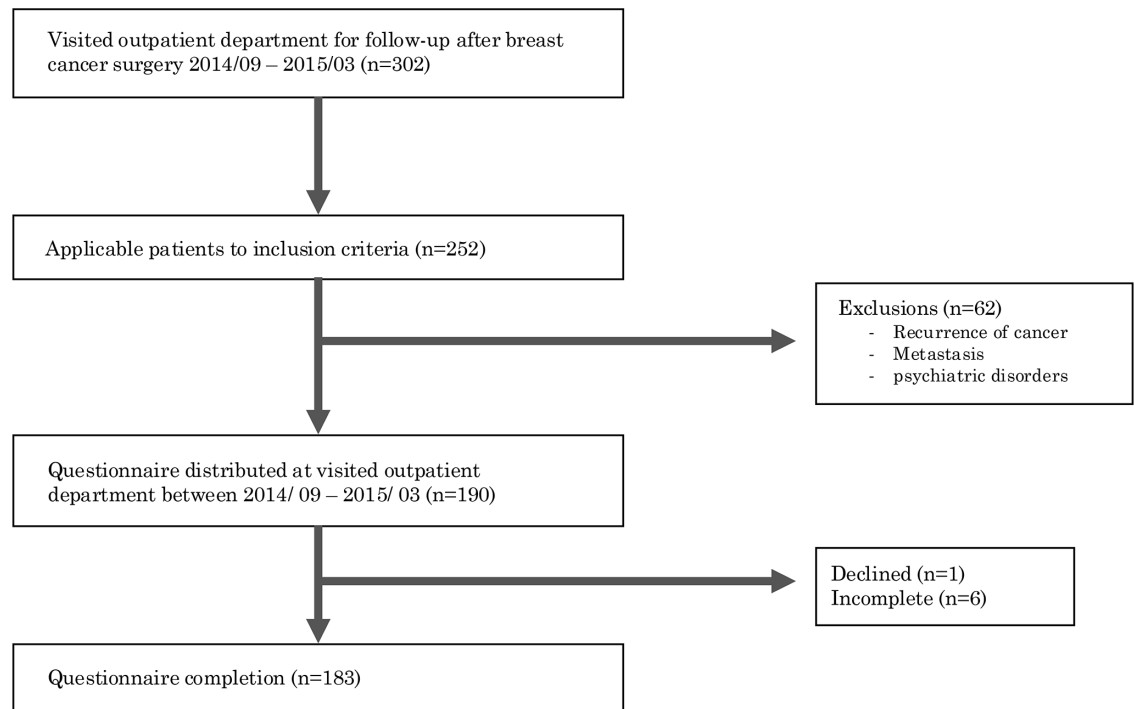

**Fig 1. Flowchart of patient inclusion process.**

## Primary outcome

In the univariate analysis, no anesthetic or non-anesthetic perioperative factors were statistically associated with PPBCS with any pain (Table 2). Similarly, no significant associations were found between the incidence of PPBCS and perioperative factors in the multivariate analysis (Table 3).

## Secondary outcomes

No anesthetic factors were significantly associated with the incidence of PPBCS with moderate-to-severe pain in the univariate analysis (Table 4). In the 'moderate-to-severe' pain group, univariate and multivariate analyses found that ALND was significantly associated with the incidence of PPBCS (odds ratio [OR], 1.99; 95% confidence interval [CI], 1.03–3.85; P = 0.041; and OR, 2.04; CI, 1.04–4.00, respectively] (Table 4 and 5). Although we observed a tendency for pain severity to decrease with the PECS block and increase with ALND, we found no factors to be significantly associated with the severity of PPBCS in either the univariate or multivariate analyses (Table 6 and 7).

**Table 1. Demographic and Patient characteristics (N = 183).**

| Demographic and Patient Characteristics | Overall (n = 183) | no or mild pain (n = 115) | moderate to severe pain (n = 68) | P value |
|---|---|---|---|---|
| BMI, median (IQR) [kg/m$^2$]* | 21.5 (19.8, 23.8) | 21.1 (19.9, 23.5) | 21.8 (19.7, 24.1) | 0.47 |
| Time since surgery, median (IQR) [mo]* | 18.1 (12.1, 24.1) | 18.9 (12.2, 24.2) | 16.8 (11.9, 23.6) | 0.35 |
| ASA score, n (%)** | | | | |
| 1 | 91 (49.7) | 57 (49.6) | 34 (50.0) | 0.96 |
| 2 | 92 (50.3) | 58 (50.4) | 34 (50.0) | |
| Type of surgery, n(%)** | | | | |
| Bp | 128 (70.0) | 83 (72.2) | 45 (66.2) | 0.39 |
| Bt | 55 (30.0) | 32 (27.8) | 23 (33.8) | |
| Type of lymph node procedure, n (%)** | | | | |
| SNB | 132 (72.1) | 89 (77.4) | 43 (63.2) | 0.039 |
| ALND | 51 (27.9) | 26 (22.6) | 25 (36.8) | |
| Operation time, median (IQR) [min]* | 103.0 (88.0, 122.0) | 102.0 (89.0, 118.0) | 106.0 (87.0, 128.0) | 0.25 |
| Anesthesia time, median (IQR) [min]* | 163.0 (146.0, 183.0) | 163.0 (146.0, 182.0) | 164.5 (142.5, 187.5) | 0.67 |
| Anesthetic maintenance, n (%)** | | | | |
| Volatile anesthesia (Sevoflurane or Desflurane) | 108 (59.0) | 68 (59,1) | 40 (58.8) | 0.97 |
| TIVA (Propofole) | 75 (41.0) | 47 (40.9) | 28 (41.2) | |
| Total dose of fentanyl standardized by BW, median (IQR) [µg/kg]* | 4.0 (3.1, 5.1) | 4.0 (3.1, 5.3) | 4.2 (3.3, 5.0) | 0.81 |
| PECS block, n (%)*** | 9 (4.9) | 7 (6.1) | 2 (2.9) | 0.49 |
| NSAIDs or acetaminophen use during surgery, n (%)** | 162 (88.5) | 103 (89.6) | 59 (86.8) | 0.57 |
| Preoperative analgesic use, n (%)*** | 6 (3.3) | 3 (2.6) | 3 (4.4) | 0.67 |
| Adjuvant chemotherapy, n (%)** | 58 (26.2) | 26 (22.6) | 22 (32.4) | 0.15 |
| Neoadjuvant chemotherapy, n (%)** | 61 (33.3) | 37 (32.2) | 24 (35.3) | 0.67 |
| Adjunvant radiotherapy, n (%)** | 141 (77.1) | 89 (77.4) | 52 (76.5) | 0.89 |
| Endocrine therapy, n (%)** | 135 (73.8) | 83 (72.2) | 52 (76.5) | 0.52 |

BMI = body mass index; Bp = partial mastectomy; Bt = total mastectomy; SNB = sentinel node biopsy; ALND = axillary lymph nodes dissection; PECS = pectoral nerves; NSAIDs = non-steroidal anti-inflammatory drugs; IQR = interquartile range; BW = body weight.

'no or mild pain' indicates 0–2 of NRS; 'moderate to severe pain' indicates 3 or more of NRS.

* Wilcoxon test; ** Chi-square test, *** Fisher's exact test.

**Table 2. Univariate analysis of association between perioperative factors and the incidence of PPBCS in any pain.**

| Variables | Univariate Analysis | |
|---|---|---|
| | any pain (n = 127) | |
| | OR (95% CI) | P value |
| Anesthetic Factors | | |
| Anesthesia time [hr] | 1.09 (0.66, 1.80) | 0.74 |
| TIVA | 0.99 (0.52, 1.88) | 0.99 |
| Total dose of fentanyl standardized by BW[μg/kg] | 0.97 (0.81, 1.15) | 0.68 |
| Infusion rate of remifentanil [0.1γ increments] | 0.98 (0.78, 1.23) | 0.85 |
| PECS block | 0.88 (0.21, 3.64) | 0.86 |
| Intraoperative analgegic use (NSAIDs or acetaminophen) | 0.90 (0.33, 2.44) | 0.83 |
| Surgical Factors | | |
| Operation time [hr] | 1.46 (0.77, 2.76) | 0.25 |
| Total mastectomy | 0.70 (0.27, 1.82) | 0.47 |
| ALND | 0.95 (0.47, 1.91) | 0.89 |
| Other Factors | | |
| Age [y] (10-y increments) | 0.76 (0.54, 1.09) | 0.14 |
| Obesity (BMI ≥ 25) | 1.09 (0.47, 2.56) | 0.84 |
| Time since surgery [mo] | 1.00 (0.95, 1.04) | 0.83 |
| Preoperative analgesic use | 2.25 (0.26, 19.8) | 0.46 |
| Chemotherapy | 0.91 (0.68, 1.23) | 0.55 |
| Radiotherapy | 2.03 (0.99, 4.16) | 0.052 |
| Endocrine therapy | 1.19 (0.59, 2.40) | 0.63 |

CI = confidence interval; TIVA = total intravenous anesthesia; PECS = pectoral nerves; NSAIDs = non-steroidal anti-inflammatory drugs; ALND = axillary lymph nodes dissection; PPBCS = persistent pain after breast cancer surgery.

any pain: NRS ≥ 1.

**Table 3. Multivariate analysis of association between perioperative factors and the incidence of PPBCS in any pain.**

| Variables | Multivariate Analysis | |
|---|---|---|
| | any pain (n = 127) | |
| | adjusted OR (95% CI) | P value |
| Anesthetic Factors | | |
| TIVA | 0.94 (0.48, 1.84) | 0.85 |
| Total dose of fentanyl standardized by BW [μg/kg] | 0.96 (0.81, 1.15) | 0.67 |
| Infusion rate of remifentanil [γ] (0.1γ increments) | 0.96 (0.76, 1.21) | 0.72 |
| PECS block | 0.80 (0.17, 3.76) | 0.78 |
| Surgical Factor | | |
| ALND | 0.96 (0.47, 1.96) | 0.91 |
| Other Factors | | |
| Age [yr] (10-yr increments) | 0.75 (0.52, 1.09) | 0.13 |
| Time since surgery [mo] | 1.00 (0.95, 1.05) | 0.96 |

CI = confidence interval; TIVA = total intravenous anesthesia; PECS = pectoral nerves; ALND = axillary lymph nodes dissection; PPBCS = persistent pain after breast cancer surgery.

any pain: NRS ≥ 1.

**Table 4. Univariate analysis of association between perioperative factors and the incidence of PPBCS in moderate to severe pain.**

| | Univariate Analysis | |
| | moderate to severe pain (n = 68) | |
| Variables | OR (95% CI) | P value |
| --- | --- | --- |
| Anesthetic Factors | | |
| Anesthesia time [hr] | 1.20 (0.75, 1.92) | 0.44 |
| TIVA | 1.01 (0.55, 1.86) | 0.97 |
| Total dose of fentanyl standardized by BW[μg/kg] | 0.97 (0.82, 1.15) | 0.76 |
| Infusion rate of remifentanil [0.1γ increments] | 0.89 (0.67, 1.16) | 0.39 |
| PECS block | 0.47 (0.09, 2.31) | 0.35 |
| Intraoperative analgegic use (NSAIDs or acetaminophen) | 0.76 (0.30, 1.91) | 0.57 |
| Surgical Factors | | |
| Operation time [hr] | 1.62 (0.92, 2.84) | 0.10 |
| Total mastectomy | 1.13 (0.81, 1.58) | 0.47 |
| ALND | 1.99 (1.03, 3.85) | 0.041 |
| Other Factors | | |
| Age [y] (10-y increments) | 0.93 (0.67, 1.29) | 0.65 |
| Obesity (BMI ≥ 25) | 1.75 (0.80, 3.81) | 0.16 |
| Time since surgery [mo] | 0.98 (0.94, 1.02) | 0.34 |
| Preoperative analgesic use | 1.72 (0.34, 8.79) | 0.51 |
| Chemotherapy | 1.41 (0.77, 2.58) | 0.26 |
| Radiotherapy | 0.95 (0.47, 1.93) | 0.89 |
| Endocrine therapy | 1.25 (0.63, 2.51) | 0.52 |

CI = confidence interval; TIVA = total intravenous anesthesia; PECS = pectoral nerves; NSAIDs = non-steroidal anti-inflammatory drugs; ALND = axillary lymph nodes dissection; PPBCS = persistent pain after breast cancer surgery.

moderate to severe pain: NRS ≥ 3.

**Table 5. Multivariate analysis of association between perioperative factors and the incidence of PPBCS in moderate to severe pain.**

| | Multivariate Analysis | |
| | moderate to severe pain (n = 68) | |
| Variables | adjusted OR (95% CI) | P value |
| --- | --- | --- |
| Anesthetic Factors | | |
| TIVA | 1.22 (0.63, 2.37) | 0.55 |
| Total dose of fentanyl standardized by BW [μg/kg] | 0.98 (0.82, 1.16) | 0.80 |
| Infusion rate of remifentanil [γ] (0.1γ increments) | 0.85 (0.61, 1.18) | 0.34 |
| PECS block | 0.32 (0.06, 1.79) | 0.20 |
| Surgical Factor | | |
| ALND | 2.04 (1.04, 4.00) | 0.038 |
| Other Factors | | |
| Age [yr] (10-yr increments) | 0.92 (0.65, 1.31) | 0.66 |
| Time since surgery [mo] | 0.97 (0.92, 1.02) | 0.19 |

CI = confidence interval; TIVA = total intravenous anesthesia; PECS = pectoral nerves; ALND = axillary lymph nodes dissection; PPBCS = persistent pain after breast cancer surgery.

moderate to severe pain: NRS ≥ 3.

**Table 6. Univariate analysis of association between perioperative factors and the severity of PPBCS.**

| Variables | Univariate Analysis | |
|---|---|---|
| | β (95% CI) | P value |
| Anesthetic Factors | | |
| Anesthesia time [hr] | −0.01 (−0.46, 0.45) | 0.98 |
| TIVA | 0.06 (−0.52, 0.65) | 0.83 |
| Total dose of fentanyl [μg/kg] | −0.10 (−0.26, 0.06) | 0.22 |
| Infusion rate of remifentanil [0.1γ increments] | −0.07 (−0.28, 0.14) | 0.52 |
| PECS block | −0.75 (−2.08, 0.57) | 0.26 |
| Intraoperative analgegic use (NSAIDs or acetaminophen) | 0.03 (−0.87, 0.93) | 0.95 |
| Surgical Factors | | |
| Operation time [hr] | 0.28 (−0.26, 0.82) | 0.31 |
| Total mastectomy | 0.01 (−0.32, 0.33) | 0.97 |
| ALND | 0.48 (−0.16, 1.11) | 0.14 |
| Other Factors | | |
| Age [y] (10-y increments) | 0.93 (0.67, 1.29) | 0.65 |
| Time since surgery [mo] | −0.01 (−0.05, 0.03) | 0.73 |
| Preoperative analgesic use | 0.12 (−1.49, 1.73) | 0.88 |
| Chemotherapy | 0.21 (−0.36, 0.79) | 0.47 |
| Radiotherapy | 0.19 (−0.50, 0.87) | 0.59 |
| Endocrine therapy | 0.22 (−0.43, 0.87) | 0.50 |

CI = confidence interval; TIVA = total intravenous anesthesia; PECS = pectoral nerves; ALND = axillary lymph nodes dissection; PPBCS = persistent pain after breast cancer surgery.

**Table 7. Multivariate analysis of association between perioperative factors and the severity of PPBCS.**

| Variables | Multivariate Analysis | |
|---|---|---|
| | β (95% CI) | P value |
| Anesthetic Factors | | |
| TIVA | 0.19 (−0.43, 0.80) | 0.55 |
| Total dose of fentanyl standardized by BW [μg/kg] | −0.10 (−0.26, 0.06) | 0.21 |
| Infusion rate of remifentanil [γ] (0.1γ increments) | −0.10 (−0.32, 0.12) | 0.37 |
| PECS block | −0.95 (−2.37, 0.47) | 0.19 |
| Surgical Factor | | |
| ALND | 0.46 (−0.18, 1.11) | 0.16 |
| Other Factors | | |
| Age [yr] (10-yr increments) | −0.10 (−0.43, 0.22) | 0.53 |
| Time since surgery [mo] | −0.01 (−0.06, 0.03) | 0.56 |

CI=confidence interval; TIVA= total intravenous anesthesia; PECS=pectoral nerves; ALND= axillary lymph nodes dissection.

## Discussion

This study found no significant associations between the occurrence of PPBCS and anesthetic management factors, nor were any associations observed between PPBCS severity and anesthetic management factors.

## Maintenance agents

Experimental studies have suggested that both propofol and volatile anesthetic agents suppress the propagation of nociceptive signals [20,21]. Propofol has been suggested to have potential protective effects against persistent postoperative pain through its anti-inflammatory, antioxidant, and neuroprotective properties, particularly at clinically relevant concentrations [22]. Also, propofol prevents remifentanil-induced hyperalgesia and exhibits analgesic properties with reduced hyperalgesia in experimental studies [22]. However, conflicting results have been reported regarding the correlation between anesthetics and PPBCS. Cho et al. [11] reported that the incidence of PPBCS was lower in patients who received TIVA than in those who received volatile anesthesia, while Lefebvre-Kuntz et al. [12] suggested that there are no effects of anesthetic agents on PPBCS. These inconsistencies may be due to the concentrations administered. Low concentrations of propofol may induce chronic pain by potentiating ionotropic purinergic P2X4 receptors, which trigger sequential signaling pathways [15]. Meanwhile, very low concentrations of volatile anesthetics were found to cause hyperalgesia and enhance pain perception during the recovery from anesthesia in an animal study [23]. In the present study, however, no association was found between PPBCS and anesthetic agents. The quantities of anesthetic agents used in clinical practice and the use of balanced anesthesia in combination with opioids may not, therefore, affect the occurrence and severity of chronic postoperative pain.

## Opioid analgesia

There were no significant associations between PPBCS and the intraoperative doses of opioids (fentanyl and remifentanil) used in this study. Preemptive analgesia has been thought to decrease acute postoperative pain and subsequently may reduce persistent postsurgical pain (PPSP) [24,25]. In contrast, intraoperative opioid use can induce acute postoperative pain through OIH. Several studies have demonstrated that a high dose of remifentanil changes the threshold for acute postoperative pain [26,27], which may subsequently transition to chronic pain. Furthermore, the combination of remifentanil with maintenance anesthetic agents may exacerbate PPSP [11,28]. This study, however, did not provide evidence that intraoperative opioid use causes persistent pain. The median remifentanil dose in our study was 0.15 μg/kg/min (IQR: 0.15–0.2), with 75% of patients receiving doses below 0.2 μg/kg/min. Previous research suggests that remifentanil doses of at least 0.2 μg/kg/min are required to demonstrate significant associations with opioid-induced hyperalgesia and persistent postoperative pain [29]. This relatively low opioid exposure in the majority of our cohort may explain the lack of association between intraoperative remifentanil use and PPBCS development and limits our ability to draw definitive conclusions about the role of intraoperative opioids in persistent pain. Additionally, our study design, with most patients receiving opioids and no opioid-free control group, does not allow us to definitively determine whether intraoperative opioid use causes persistent pain. As there are limited reports investigating the correlation between high doses of remifentanil and PPSP [13,14], further studies are needed to elucidate this relationship.

## Non-opioid analgesia

Although multimodal analgesia has been proposed to be beneficial in the prevention of PPSP [20], the statistical significance of non-opioid analgesics in PPSP remains unclear. The administration of NSAIDs or acetaminophen, rather than opioids, during surgery to reduce postoperative pain would relieve acute postoperative pain while inhibiting OIH [21], is, therefore, expected to inhibit subsequent PPSP. However, the effects of these non-opioid analgesics on the prevention of PPSP are yet to be elucidated [30]. The present study showed that the intraoperative use of these analgesics had no significant effect on the prevalence and intensity of chronic pain.

## Use of local anesthetics and regional anesthesia

Regional anesthesia appears to benefit the occurrence and severity of PPBCS [8]. Recently, the PECS II block, an interfacial nerve block for the breast and axillary regions, has been found to be effective in reducing acute postoperative

pain with minimal complications [31]. However, only few randomized controlled trials (RCTs) have examined the association between the PECS block and PPBCS, and the relevant literature presents conflicting results [9,10]. All PECS blocks performed in this study were PECS II blocks, utilizing 20 ml of 0.25% ropivacaine. The blocks were performed by a limited number of anesthesiologists with expertise in regional anesthesia techniques. The PECS II block was performed under ultrasound guidance with injections between the pectoralis major and minor muscles (PECS I component) and between the pectoralis minor and serratus anterior muscles (PECS II component). In our study, the limited number of patients receiving PECS block (n = 9) prevents us from drawing definitive conclusions about its effect on PPBCS. Further adequately powered studies are needed to investigate this potential relationship.

## Other factors

In general, younger age and ALND are recognized risk factors of PPBCS [19,32]. Consistent with previous reports, the present study showed that ALND is significantly associated with the incidence of PPBCS; however, age did not show statistically significant correlation. The presence of unexamined confounding factors, such as the preservation of the intercostobrachial nerves and potential adjuvant treatments, could have suppressed the incidence of PPBCS.

## Prevalence

The definition of PPBCS, based on pain intensity and assessed using an NRS or visual analog scale, varies among studies. In this study, the prevalence of PPBCS of any intensity was higher than that previously reported [3–5,33]. When compared to the data on supplementary appendix presented in the systematic review by Wang et al., no factors were found to contribute to the unusually high prevalence in the present study. The following are the potential reasons for high prevalence of PPBCS in this study. First, it is possible that the incidence is higher in Japanese and Asian populations. However, no such evidence has been established to date, requiring more cross-racial data to be accumulated to clarify. Second, the questionnaire collection method is a possible factor. A systematic review indicated that self-reported patients had a significantly higher prevalence of PPBCS than those assessed by a clinician (36% [95% CI, 33–40%] vs. 23% [95% CI, 12–35%], respectively) [5]. In this study, the method used to obtain questionnaires was to have patients fill out the questionnaires directly in the outpatient clinic. It is possible that the responses were timely for the patients and that potential patients with mild pain could have been picked up. Therefore, our method of distributing the questionnaire might be useful for identifying patients with PPBCS. However, further studies are needed to clearly define PPBCS based on pain intensity and to establish an optimal method for evaluating PPBCS.

## Strengths and limitations

Although the results of this study may not be novel compared to those previously reported, the high response rate to the questionnaire (96.3%) is an advantage over previous reports. Most studies evaluating PPBCS using questionnaires have been conducted by mail, email, or telephone, with response rates ranging from 57% to 83%. The incidence of PPBCS in this study was approximately 70%, which was higher than that reported previously (35% [95% CI, 32–39]). The higher incidence rate observed in our study may be related, therefore, to the higher response rate. The questionnaires were provided directly to participants by medical doctors during postoperative follow-up visits, and participants were asked to place them in a box before leaving the hospital. This system seemed to contribute to the higher response rate, providing accurate data for studying the effects of anesthetic management on PPBCS.

However, this study also had several important limitations that should be considered when interpreting the results. First, the retrospective observational design inherently limits the ability to establish causality, and our findings should be interpreted with this limitation in mind. Second, the data were collected between 2012 and 2014, a period when some current

regional anesthetic techniques such as Erector Spinae Plane block (ESP) and Serratus Plane Block had not yet been formally described or incorporated into standard clinical practice. Third, no formal sample size calculation was performed, which may have affected our ability to detect significant associations for some variables. Fourth, patients with preoperative pain or other chronic pain conditions were not systematically assessed, which represents a limitation of our retrospective design and could have influenced our findings on PPBCS. Despite these limitations, our study provides valuable insights into the relationship between anesthetic management and PPBCS, particularly due to the high response rate to our questionnaire.

## Conclusion

In this study, we did not identify any anesthetic management strategies that significantly prevent PPBCS. Although the PECS block appeared to reduce the severity of PPBCS, its effect was not statistically significant. However, these findings are constrained due to the limited number of participants. Further research is needed to identify anesthetic factors that may prevent PPBCS.

## Supporting information

**S1 Fig. Flowchart of patient inclusion and exclusion process.** Detailed flowchart showing patient recruitment, inclusion/exclusion criteria application, and final study population selection from initial 302 patients to final 183 participants. (PDF)

**S1 Dataset. Complete anonymized dataset.** De-identified dataset containing all 23 variables for 183 patients used in the analysis, including demographic data, perioperative factors, anesthetic management details, and pain assessment outcomes (NRS scores). (CSV)

**S1 File. Data dictionary.** Detailed description of all 23 variables in the S1_Dataset.csv file including variable names, definitions, coding schemes, measurement units, and value ranges for the PPBCS study dataset. (DOCX)

## Author contributions

**Conceptualization:** Tomoyuki Miyazaki.

**Data curation:** Yu Sakai, Ayako Kobayashi, Mariko Akata.

**Formal analysis:** Yu Sakai, Tojo Kentaro, Takahiro Mihara.

**Funding acquisition:** Tomoyuki Miyazaki.

**Investigation:** Akimitsu Yamada, Kazutaka Narui, Sadatoshi Sugae, Nobuyasu Suganuma.

**Methodology:** Ayako Kobayashi, Mariko Akata, Akimitsu Yamada, Kazutaka Narui, Sadatoshi Sugae, Nobuyasu Suganuma.

**Project administration:** Tomoyuki Miyazaki.

**Supervision:** Takahisa Goto, Tomoyuki Miyazaki.

**Validation:** Takahiro Mihara.

**Writing – original draft:** Yu Sakai.

**Writing – review & editing:** Ayako Kobayashi, Mariko Akata, Tojo Kentaro, Takahiro Mihara, Akimitsu Yamada, Kazutaka Narui, Sadatoshi Sugae, Nobuyasu Suganuma, Takahisa Goto, Tomoyuki Miyazaki.

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
