## [Decision Letter · Decision Letter 0]

12 Apr 2025

Dear Dr. Sakai,

Thank you for submitting your manuscript to PLOS ONE. After careful consideration, we feel that it has merit but does not fully meet PLOS ONE’s publication criteria as it currently stands. Therefore, we invite you to submit a revised version of the manuscript that addresses the points raised during the review process.

Please submit an updated version after major revisions.

We look forward to receiving your revised manuscript.

Kind regards,

Mohammad Mofatteh, PhD, MPH, MSc, PGCert, BSc (Hons), MB BCh (c)

Academic Editor

PLOS ONE

Journal Requirements:

4. In the online submission form, you indicated that data cannot be shared publicly because they contain sensitive patient information. Data are available from the corresponding author (contact via ysakai61@yokohama-cu.ac.jp) for researchers who meet the criteria for access to confidential data. Access requires prior approval from the Ethics Committees of both Yokohama City University Hospital (approval number: B14050823) and Yokohama City University Medical Center (approval number: D1408008).

5. Please remove all personal information, ensure that the data shared are in accordance with participant consent, and re-upload a fully anonymized data set.

Additional Editor Comments :

Please submit an updated version after major revisions.

Reviewers' comments:

Reviewer's Responses to Questions

**Comments to the Author**

1. Is the manuscript technically sound, and do the data support the conclusions?

Reviewer #1: No

Reviewer #2: Yes

Reviewer #3: Yes

Reviewer #4: Yes

2. Has the statistical analysis been performed appropriately and rigorously?

Reviewer #1: N/A

Reviewer #2: Yes

Reviewer #3: Yes

Reviewer #4: Yes

3. Have the authors made all data underlying the findings in their manuscript fully available?

Reviewer #1: Yes

Reviewer #2: Yes

Reviewer #3: Yes

Reviewer #4: Yes

4. Is the manuscript presented in an intelligible fashion and written in standard English?

Reviewer #1: Yes

Reviewer #2: Yes

Reviewer #3: Yes

Reviewer #4: No

Reviewer #1: Whilst I appreciate the effort, this is a very small and retrospective study with all the attendant problems that accompany this format. There are potentially many confounders that could affect your results and the only way to answer your question is to perform a proper RCT with standardised perioperative analgesia. Perhaps study something that has actually been shown to reduce pain e.g. ketamine. The study was conducted more than 10 years ago which is a big delay in publication - techniques may even have changed over this time. Your perioperative opioid doses are much to low to cause OIH or hyperalgesia. Propofol, if anything, will reduce postoperative pain.

Reviewer #2: Dear Authors, thank you for the opportunity to review your study.

This study addresses an important clinical issue regarding PPBCS prevention. It investigates the association between anesthetic management and the incidence of PPBCS.

In my opinion, there are aspects that need clarification and concepts that need improvement.

- It would be appropriate to include among the study's limitations the fact that it is a retrospective and observational study, which inherently limits the ability to establish causality. (minor review).

- Data were collected in a period of time when trunk blocks were not yet taken into account as part of standard clinical practice. Many of them such as Erector Spine Plane block (ESP) and Serratus Plane Block had not even been formally described at the time. It would be appropriate to specify this into discussion. (major review)

- The classification of PPBCS into 'no,' 'mild,' and 'moderate-to-severe' pain is based on NRS scores, but the choice of cut-off values is not well justified. Clarifying the rationale for NRS cut-offs would be advisable. (major review)

- There is no mention of preoperative pain assessments or how other chronic pain conditions were accounted for. Specify if this was an exclusion criteria (minor review).

- Specify whether the study protocol was drafted in relation to the Declaration of Helsinki. (minor review).

Reviewer #3: Congratulations on this retrospective study, the manuscript is very well written and easy to follow. variability in retrospective clinical studies is hard to controlled by limitations are clearly recognized and stated. Please see some comments:

Abstract:

Line 44 and 45: I would recommend to eliminate the sentence that reads "pectoral nerve block showed a tendency to reduce...". I think the p value is far away from significance enough that this statement may be misleading. As the authors stated in the discussion the number of blocks performed in this study population was very low. Low enough that in my opinion precludes from any statistically interpretation. I would recommend to match the abstract conclusion closer to the one used in Line 232.

Introduction:

Line 56: the definition of PPBCS appears to be inconsistent or lacking throughout the literature. Could the authors define what was considered "persistent pain"in this study? Was it more than 6 months after surgery? I think it should be more clearly explained

Statistical analyses:

Line 116: How was normality assessed?

Was there a sample size calculation performed at any point during the analysis to assess which variables may be underpowered?

Discussion:

Line 242 and 243: Can you please re-write this sentence? It reads awkward

Also I think it should be "this difference may depend..."

Line 260-261: I would rephrase this statement. I do not think this study can proof or disproof that intraoperative opioid use can cause persistent pain since the majority of patients did receive opioids. There was no control opioid-free population to compare.

Line 280 and 281: I do not think you can conclude for sure that the difference was not statistically significant because of the limited number without running at the analysis with bigger numbers. It may be, but you cannot predict factors such as inter-individual variability that could make the results same or even less significant with a larger population.

Sorry, I could not assess fig 1 since it was showing black for me

Reviewer #4: The review of the manuscripts"Effects of anesthetic management on persistent pain after breast cancer surgery"

Thank you foy my opportunity of reviewing.

The author have described that the association of perioperative factors, including anesthetic management, with the subsequent incidence and intensity of PPBCS.

Although this study is interesting,but there is serious concern.

Sincerely,

Major concern:

#1:Study period

This study was conducted in 2014.

This was more than 10 years ago, and I think that technological advances such as anesthetic drugs and blocks have had a significant impact during that time. I think this is a serious limitation of the study.

2:Block

The author concluded that the PECS block might have a preventive effect on the development of PPBCS.

However, there are no details regarding the PECS block. (For example, which PECS1, PECS2, or both?, which local anesthetic used,? and who blocked?).

If information is available, it should be provided.

**Do you want your identity to be public for this peer review?** For information about this choice, including consent withdrawal, please see our Privacy Policy

Reviewer #1: No

Reviewer #2: No

Reviewer #3: No

Reviewer #4: No

---

## [Author Response · Author response to Decision Letter 1]

27 Jun 2025

Dear Dr. Mohammad Mofatteh and PLOS ONE Editorial Team,

Thank you for providing us with the opportunity to revise our manuscript titled "Effects of anesthetic management on persistent pain after breast cancer surgery." We greatly appreciate the constructive feedback from the reviewers and the editorial team, which has significantly improved the quality of our work.

Summary of Major Revisions

We have carefully addressed all comments and requirements outlined in your revision letter. Below is a comprehensive summary of the changes made:

Editorial Requirements Addressed:

1. PLOS ONE Style Compliance: We have thoroughly reviewed and reformatted our manuscript to meet all PLOS ONE style requirements, including proper file naming conventions.

2. Funding Information: We have corrected the discrepancy between the 'Funding Information' and 'Financial Disclosure' sections to ensure accurate grant number reporting.

3. Data Availability: We have prepared a fully anonymized dataset and updated our Data Availability statement. Due to ethical restrictions imposed by our institutional review boards to protect patient privacy, de-identified data are available upon request from the corresponding author, subject to ethics committee approval.

4. Data Anonymization: We have created a completely anonymized dataset with all personal identifiers removed, ensuring compliance with participant consent and data protection requirements.

Reviewer Comments Addressed:

Reviewer #1 Response:

• Enhanced discussion of retrospective study limitations

• Added clarification about the temporal gap since data collection (2012-2014)

• Provided detailed explanation of relatively low opioid doses and their implications for OIH

• Expanded discussion of propofol's potential analgesic and neuroprotective effects

Reviewer #2 Response:

• Added explicit acknowledgment of causality limitations in retrospective design

• Included historical context about regional anesthesia techniques not yet available during study period

• Provided clear justification for NRS cut-off values based on literature and clinical significance

• Added Declaration of Helsinki compliance statement

• Acknowledged limitation regarding preoperative pain assessment

Reviewer #3 Response:

• Removed misleading statement about PECS block tendency from abstract

• Added detailed methodology for normality assessment

• Acknowledged lack of formal sample size calculation

• Revised awkward sentences for clarity

• Enhanced discussion about study limitations regarding opioid conclusions

• Added comprehensive PECS block technical details

Reviewer #4 Response:

• Explicitly acknowledged the significant limitation of 10-year-old data

• Added detailed PECS block methodology including type (PECS II), local anesthetic (ropivacaine 0.25%, 20ml), technique, and practitioner qualifications

• Enhanced discussion of how anesthetic techniques have evolved since data collection

Additional Improvements Made:

• Restructured Discussion Logic (Lines 238-256): Reorganized the "Maintenance agents" section to improve logical flow by presenting experimental evidence about propofol and volatile agents' neuroprotective properties first, followed by clinical studies and their conflicting results, leading to a more coherent narrative about anesthetic agents and PPBCS.

Key Scientific Improvements:

1. Enhanced Methodology Section: Added detailed descriptions of anesthetic techniques, particularly PECS blocks, and statistical methods.

2. Strengthened Limitations Discussion: Comprehensive acknowledgment of retrospective design limitations, temporal context, sample size considerations, and missing variables.

3. Improved Clinical Context: Better integration of current literature regarding anesthetic agents, opioid dosing, and regional anesthesia techniques.

4. Clarified Conclusions: More conservative and accurate interpretation of results, particularly regarding PECS block effects and opioid relationships.

5. Restructured Discussion Logic: Reorganized the "Maintenance agents" section to improve logical flow by presenting experimental evidence first, followed by clinical studies and their conflicting results, leading to a more coherent narrative about anesthetic agents and PPBCS.

Compliance with Journal Policies

• Ethics: Study conducted in accordance with Declaration of Helsinki principles

• Data Sharing: Anonymized dataset prepared following BMJ guidelines for clinical data de-identification

• Formatting: Manuscript formatted according to PLOS ONE style templates

• References: All citations properly formatted and accessible

Files Submitted

1. Revised Manuscript (clean version without track changes)

2. Manuscript with Track Changes (showing all revisions)

3. Response to Reviewers (point-by-point responses to each reviewer)

4. Anonymized Dataset (supporting information file)

5. Cover Letter (this document)

Data Availability Statement

All relevant data are within the manuscript and its Supporting Information files. The complete de-identified dataset underlying the results presented in this study is available as Supporting Information file S1_Dataset.csv. This dataset contains all variables used in the statistical analyses, including demographic characteristics, perioperative factors, anesthetic management details, and pain outcomes for all 183 participants. No additional data are required to replicate the study findings.

Conclusion

We believe that the extensive revisions made in response to reviewer and editorial feedback have significantly strengthened our manuscript. The study provides valuable insights into the relationship between anesthetic management and persistent pain after breast cancer surgery, despite the limitations inherent in its retrospective design.

We are confident that our revised manuscript now meets the high standards of PLOS ONE and contributes meaningful evidence to the field of perioperative pain management in breast cancer surgery.

Thank you for your consideration of our revised submission. We look forward to your feedback and hopefully, acceptance for publication in PLOS ONE.

Sincerely,

Yu Sakai

---

## [Decision Letter · Decision Letter 1]

21 Sep 2025

Effects of anesthetic management on persistent pain after breast cancer surgery

PONE-D-25-08461R1

Dear Dr. Sakai,

We’re pleased to inform you that your manuscript has been judged scientifically suitable for publication and will be formally accepted for publication once it meets all outstanding technical requirements.

Kind regards,

Dr. Mohammad Mofatteh, PhD, MPH, MSc, PGCert, BSc (Hons), MB BCh BAO (c)

Academic Editor

PLOS ONE

Additional Editor Comments (optional):

Congratulations.

Reviewers' comments:

Reviewer's Responses to Questions

**Comments to the Author**

Reviewer #2: All comments have been addressed

Reviewer #3: All comments have been addressed

Reviewer #4: (No Response)

2. Is the manuscript technically sound, and do the data support the conclusions?

Reviewer #2: Yes

Reviewer #3: Yes

Reviewer #4: Yes

3. Has the statistical analysis been performed appropriately and rigorously?

Reviewer #2: Yes

Reviewer #3: Yes

Reviewer #4: Yes

4. Have the authors made all data underlying the findings in their manuscript fully available?

Reviewer #2: Yes

Reviewer #3: Yes

Reviewer #4: No

5. Is the manuscript presented in an intelligible fashion and written in standard English?

Reviewer #2: Yes

Reviewer #3: Yes

Reviewer #4: Yes

Reviewer #2: I have reviewed the revised manuscript and the authors' responses to the reviewers' comments. I am substantially satisfied with the revisions and the point-by-point responses provided. The authors have addressed the major concerns raised in my previous review.

Reviewer #3: Thanks for addressing all my comments. I think the manuscript, especially the discussion reads much better after all the edits.

Reviewer #4: Dear author,

It seems that the comments in "Word" remain(Page 13,line 224). Did you check it carefully before submitting ?

Careless resubmission may result in your submission being rejected, even if the content is fine.

So I strongly recommend that you check it carefully before submission.

**Do you want your identity to be public for this peer review?** For information about this choice, including consent withdrawal, please see our Privacy Policy

Reviewer #2: No

Reviewer #3: No

Reviewer #4: No

---

## [Editor Report · Acceptance letter]

PONE-D-25-08461R1

PLOS ONE

Dear Dr. Sakai,

I'm pleased to inform you that your manuscript has been deemed suitable for publication in PLOS ONE. Congratulations! Your manuscript is now being handed over to our production team.

Kind regards,

on behalf of

Dr. Helmar Bornemann-Cimenti

Academic Editor

PLOS ONE